# Alternative Paradigms in Animal Health Decisions: A Framework for Treating Animals Not Only as Commodities

**DOI:** 10.3390/ani12141845

**Published:** 2022-07-20

**Authors:** Noguera Z. Liz Paola, Paul R. Torgerson, Sonja Hartnack

**Affiliations:** 1Section of Epidemiology, Vetsuisse Faculty, University of Zürich, 8057 Zürich, Switzerland; ptorgerson@vetclinics.uzh.ch (P.R.T.); sonja.hartnack@access.uzh.ch (S.H.); 2Epidemiology and Biostatistics, Life Science Zurich Graduate School, University of Zurich, 8057 Zurich, Switzerland

**Keywords:** zoonosis, animal health, environmental ethics, One Health, framework

## Abstract

**Simple Summary:**

Since zoonotic diseases can be transmitted from animals to humans, more comprehensive measures are needed when preventing and controlling these diseases. Because the value of animals is mainly based on monetary terms, animals are typically treated as commodities, impacting public health decisions. Therefore, a framework is proposed to value the health of animals beyond money for public health decision-making with a “One Health” approach. The aim is to have more comprehensive animal values based on the opinion of societies. However, tackling the dilemmas related to animal diseases, public health, and welfare still represents a challenge and a work in progress.

**Abstract:**

Zoonoses are diseases transmitted from (vertebrate) animals to humans in the environment. The control and prevention of these diseases require an appropriate way to measure health value for prudent and well-balanced decisions in public health, production costs, and market values. Currently, the impact of diseases and animal disease control measures are typically assessed in monetary values, thus lacking consideration of other values such as emotional, societal, ecological, among others. Therefore, a framework is proposed that aims to explore, understand, and open up a conversation about the non-monetary value of animals through environmental and normative ethics. This method might help us complement the existing metrics in health, which are currently DALY and zDALY, adding more comprehensive values for animal and human health to the “One Health” approach. As an example of this framework application, participants can choose what they are willing to give in exchange for curing an animal in hypothetical scenarios selecting a human health condition to suffer, the amount of money, and lifetime as a tradeoff. Considering animals beyond their monetary value in public health decisions might contribute to a more rigorous assessment of the burden of zoonotic diseases, among other health decisions. This study is structured as follows: after a brief introduction of zoonoses, animal health, and health metrics, briefly, different environmental health perspectives are presented. Based on this, a framework for animal health decisions is proposed. This framework introduces the “anthropozoocentric interface” based on anthropocentrism and zoocentrism perspectives.

## 1. Introduction

Approximately 75% of emerging human pathogens originate from animals [1]. Certain diseases that affect the human population are considered “human diseases,” despite having an animal origin, such as HIV, dengue, and others [2]. Animal diseases that can be transmitted to humans are called “zoonoses,” and their transmission can occur directly and indirectly through food, air, water, vectors, and fomites. Since zoonoses imply multiple factors and more than one species, more challenges are encountered to control and prevent the spread of these diseases.

## 2. Background

There is ample evidence that several factors contribute to the risk of zoonoses. Zoonosis outbreaks are not only due to the influence of weather and climate change [3] but mainly due to the influence of anthropogenic impacts on the environment [4,5], such as overpopulation, overconsumption [6], deforestation, biodiversity loss [7], pollution of natural resources (air, water, and soil), intensification of animal and plant trade [8], civil unrest, war [9], and famine [10]. The outbreak scenarios repeat over time with different infectious diseases [11], and the frequency of outbreaks could increase if appropriate measures are not taken. For that reason, zoonosis control requires intervention in the transmission pathway between animals and humans to prevent diseases, e.g., milk pasteurization, vaccination, habitat conservation, and more sustainable alternatives. Due to this, environmental ethics, a branch of applied philosophy, has lately gained more attention in developing solutions in an integrative approach–even though its beginnings were in the 1960s [12].

Animal health mainly matters when it represents an economic loss, e.g., governments spend more resources on diseases in livestock production, such as the control of foot-and-mouth diseases, than on wildlife. In Europe, the foot-and-mouth disease directive 2003/85/EC mentions the relevance of economic aspects but also refers to ethical aspects by “…increasing the profitability of livestock farming and facilitating trade in animals and animal products. At the same time the Community is also a Community of values, and its policies to combat animal diseases must not be based purely on commercial interests but must also take genuine account of ethical principles” [13]. However, there are no specific guidelines on how to include ethical principles in animal disease control.

Animal economic value has been the main component of decisions and actions of international organizations that regulate health and trade, such as the World Organisation for Animal Health (WOAH, previously the OIE) and the World Trade Organization (WTO). Nonetheless, these organizations have lately raised more awareness to address health challenges in an integrative approach (Figure 1), such as the One Health High-Level Expert Panel (OHHLEP) in 2020 [14] and the World Trade Organization (WTO) Agreement on the Application of Sanitary and Phytosanitary Measures (SPS Agreement), in 1995, to protect human, animal, and plant health—because trade and health go together [15]. In a socio-ecological system, the economy is not only about economic growth, but it is also a way of delivering well-being to populations through the protection of humans, plants, and animals. An example of this is “The 2030 Agenda for Sustainable Development” [16]—an action plan adopted by all United Nations Member States in 2015 for the planet, people, and prosperity—and the Wellbeing economy alliance (WEAll)—a 10-year project that aims to transform the economic system to provide social justice on a healthy planet [17]. However, tangible actions are still pending to contribute to the goals of these international collaborations. 

Certain metrics have been created to measure population health in non-monetary terms. For human health, there are well-known and accepted metrics to identify the health priorities in populations for decision-making in global health policy implemented through the World Health Organization (WHO) [18,19,20]. These metrics exist because money is not an equitable means of valuing human life and health, e.g., a high-income country, that logically spends more money on health, may have a greater financial burden of ill health despite a lower incidence of disease than a low-income country. One of the metrics created to solve this problem, making the measurement of diseases more equitable, is the *Disability Adjusted Life Years* (DALY) metric. The DALY metric consists of the *years of life lost* (YLL) due to early mortality and the *years lost due to disability* (YLD) caused by a disease or condition. The YLD metric depends on the disability weight assigned to the health condition. The disability weight indicates the level of severity of a health condition and can be calculated by several techniques such as pairwise comparison, time trade-off, and visual analogue scale [21].

Regarding animal health, the existing metrics for decision making in public health have, so far, mainly been based on monetary values. Nevertheless, the DALY has been modified to estimate the burden of zoonotic diseases considering the animal loss based on time trade-off (i.e., time taken to earn sufficient money to “replace” the value of the animal). This metric is called *zoonosis Disability Adjusted Life Years*, or zDALY [22]. However, zDALY has not yet included other factors (emotional attachment, cultural beliefs, or intrinsic value). Such factors need to be included in metrics when valuing animal diseases and animal health in order to consider more species (in addition to livestock) and avoid their underestimation—see Figure 2.

Money is the most accepted commodity by general consent as a medium of economic exchange [23], which also applies to the value of animals. Money is also convenient for weighing different interests that affect public decisions. However, thinking just in terms of money for public health decisions can affect how populations deal with zoonoses, food safety, antibiotic resistance, sustainable development, and welfare. Thus, decision making based on just monetary equivalents bears serious ethical concerns and calls for alternative practices.

## 3. Framework on Alternative Paradigms in Animal Health Decisions

Animal heterogeneity requires a different and more comprehensive approach to assessing animal value and health. This framework considers that the value of animals differs by species, among cultures, beliefs, place, time, and context, as well as personal needs, wishes, and expectations (Figure 3). To better understand how people can value health and animals in a more integrative approach, four main perspectives of environmental ethics are considered: anthropocentrism, zoocentrism, biocentrism, and ecocentrism [12] (see Figure 4). 

Anthropocentrism believes that humans are the only beings with moral standing [24]. In contrast, zoocentrism assumes that at least some animals, including humans, have moral standing [25]. Within zoocentrism exists a moral point of view called “pathocentrism”, which recognizes animal suffering as morally significant [25]. Certain countries have changed their laws to start treating their animals with dignity, such as Switzerland (since 1992) [26] and Spain (since 2022) [27], among others [28]. From the biocentric perspective, all living beings have an intrinsic value (including humans, non-human animals, plants), but that does not imply that all of them have an equal value [29].

For ecocentrism, the whole nature has an intrinsic value, including living beings and non-living things [30]. However, ecocentrism does not strictly imply an equivalent value for all living beings and non-living things but protection of natural resources to ensure wellbeing, and sustainability [31,32]. For example, the New Zealand government recognized the Whanganui River as a legal person in order to not only protect the river but also to respect the Māori people who have an ancestral connection with that river [33]. In this framework, environmental ethics intertwine with normative ethics, the latter being the ways in which humans behave and treat others (including animals). The main theories of normative ethics consist of consequentialism (consequence of actions) or utilitarianism (the greatest amount of good for the greatest number of people), deontology (right-based: duties, rules), and virtue ethics (moral character) [34,35,36]. It has been considered that mainly consequentialism and deontology ethics played a role in the intrinsic value of animals [37,38]. Under the scope of consequentialism, we find utilitarianism, which maximizes well-being [23]. According to Nussbaum, among all ethical theories in normative ethics, utilitarianism has contributed the most to the recognition of the intrinsic value of animals [39].

### Anthropozoocentric Interface

The increasing interest that society has in animals is known as the “animal turn” [40]. The animal turn represents a change in mindset which has not yet taken place in public health decisions. This framework introduces the “anthropozoocentric interface” in order to explore a “mindset shift” or, at least, a “scout mindset” [41] regarding the value of animals’ health in an anthropocentric and zoocentric interface. The anthropozoocentric interface arises from the combination of anthropocentrism [24] and zoocentrism [25]—see (Figure 5). This means that we, as humans, are not restricted to only one perspective for our decisions on health and animals. Consequently, according to the situation, the “anthropozoocentric interface” has flexible boundaries where the anthropocentric and zoocentric perspectives can be shifted from one to another. According to the perspective that is chosen, the way that humans value animals influences how they make decisions about animal health and welfare [42,43,44,45]. Therefore, in the “anthropozoocentric interface,” the opinions and perspectives are flexible.

Regarding the inclusion of non-monetary value of animal health, this framework proposes a method through reflexive questions in an “anthropozoocentric” interface. So, how do we value animal health beyond money (not only as commodities)? The level of importance of essential factors, such as health, time, and money, varies according to each person’s priorities [46,47,48,49]. What if we must give in exchange part of our health, money, and lifetime to cure a sick animal? Answering these questions will help identify how we perceive and value animals and how we make decisions about them—considering that we share with animals not only the environment but also emotions and potential diseases. All decisions about animals’ health affect us directly or indirectly. For this reason, science and society need to collaborate for better and fairer decisions in health and welfare, not only in empirical aspects but also normative ones. 

This framework aims to explore updated perceptions about animals in order to improve existing animal and human health metrics for zoonoses. These perceptions included in metrics can be applied to laws based on evidence in the long term. Health, sickness, pain, and death are “comparable” benchmarks for humans and non-human animals because we all can suffer from diseases, pain, and our life cycle ends with death. Therefore, the first challenge is asking ourselves if we are ready to compare the value of animal health to our own health, thinking out of speciesism—this being an anthropocentric perspective which consists of the discrimination against other species different from ours [50]. A perspective beyond anthropocentrism does not mean that we aim to value animals equally. However, it is up to people to decide how they value the pain, suffering, and sickness of animals, how important it is for them to avoid them, and at what cost. In this respect, this framework aims to estimate the disability weight (DW) of animal diseases in a pairwise comparison with known DWs of human conditions using similar methodology to the Global Burden of Diseases (GBD) studies [51].

Extreme ways of thinking are not ideal for the health of populations; thus, prudent decisions and actions in health are needed. 

An example of application is shown in Figure 6.

This framework not only calls for quantitative but mainly for qualitative analysis as a starting point. Diverse data sources (surveys, open societal questions) need to be included through a culturally and context-sensitive method that does not judge any cultural belief as right or wrong, better or worse. As public health concerns everyone, participants with or without animals in charge deserve to share their opinion about animal health since, directly and indirectly, this affects all of us.

A modified GBD methodology is proposed in this framework for animal health valuation to determine which health state in an animal causes equivalent suffering to a person, such as an owner or not. 

The following points consist of an example to explore whether the burden of animal diseases and injuries can be estimated and directly incorporated into the DALY metric through the morbidity suffered by people in an “anthropozoocentric” interface (beyond its monetary value): (1)Compare an animals’ health condition to human health conditions. This would give us an “exchange rate” between the disability weights given to human diseases and those of animals;(2)Elicit how much money people would be willing to pay to cure an animals’ disease;(3)Elicit how much time of their lives people would be willing to trade to cure an animals’ disease.

Based on the *Population Health Equivalence* (PHE) [52], this can be used to add to or modify the AHE in the zDALY, i.e., the “*Animal Health Equivalence*” (AHE). For the GBD, the PHE was made to compare hypothetical health programs. In contrast, the AHE covers hypothetical animal scenarios where humans have a whole severity range of human diseases to choose from in exchange for curing an animal under specific conditions, alternatively, the willingness to pay (which can then be converted to a time trade-off) and direct time trade-off (how much time you would give up for your animal). Therefore, the values for animal health conditions can be anchored based on the answers of participants. This is equivalent to and would replace the AHE in the zDALY.

Regarding the time-trade off, a modification of the original metric is suggested from “*Compensating Variation for a Health Gain* (CVG)” [53] to “*Compensating Variation for an Animal Health Gain* (CVAG)” (Figure 7A,B), which would be an alternative method to estimate the AHE in the zDALY.

The proposed framework can complement the existing metric, adding more comprehensive values for animal and human health in order not to only focus on its monetary value or just on livestock.

## 4. Discussion

This framework approaches different and more comprehensive ways of assessing the health and welfare of the animal and human populations for zoonoses decision making. Integrating this complexity is a challenge due to the diverse factors and perceptions of animals. Animal topics trigger controversies mainly when discussing their value, which impacts their health and quality of life. The ways in which humans value animals are dynamic and relative due to the conflation of several factors. This can reveal part of “ourselves,” based on what we respect, believe, know, want, and how we feel about it. For example, feminists have contributed to animal protection throughout their history because they have felt the need to protect oppressed populations such as animals [54]. Most feminists have sympathy for animals because, in the past, women were treated as objects as well as animals—and in certain countries, they still are.

How humans perceive animals and their value more likely depends on how they interact. The human–animal relationship (Figure 8) has always been a complex and dynamic process in diverse ways and forms, according to time and place. This human–animal interaction has changed due to different reasons such as sources of food, transportation, companionship, service animals (guide dogs, landmine detector rats), therapy [55], ornament, animals as source of inspiration (being part of superstitions, legends, myths, paintings, sculptures, and biomimetics [56]), to express identities [40] (in some indigenous cultures or as a representation of specific groups in society), and as being a part of history [57] and our story. In some instances, the human–animal interaction has been critical lately [58] because of conflicts between farmers and wild animals due to habitat and resource competition (human encroachment); in other cases, the increased anthropomorphism that humans attribute to their pets to consider them as children.

The context or situation also influences the value of animals, e.g., a pig as a pet is not perceived the same as a farm pig or as a laboratory pig. Regarding wildlife species, their value depends mainly on their population size, their role in the ecosystem, and whether or not they are native to a specific place or ecosystem because that implies the level of damage or benefit that such interactions with their environment can cause. For example, the value of beavers in Tierra del Fuego is perceived as completely opposite to their value in Canada. The reason is that beavers are not native to South America; thus, they are destroying part of the ecosystem in Tierra del Fuego and invading more areas in Argentina and Chile [59]. In contrast, beavers in Canada are a national symbol and a native species, so their impact is regulated by all the components of the natural environment.

Every species is different, and logically, humans have different perceptions about them. Many humans have at least another species to which they feel strongly connected, such as pets [60,61,62,63,64]. In some cases, pets are preferred over other humans, even partners or children—as several online polls have already shown it. According to Walsh, 57% of participants in a survey would choose their pet if stranded on a desert island with only one companion [62]. However, certain animal-related polls and surveys tend to be slightly restrictive and biased since animal lovers participate the most. 

As some species are preferred, others cause aversion or phobia, such as snakes [65,66], spiders [67,68,69], rats [70], cockroaches [71], bats [72,73,74]—among the most popular. When such aversion or phobia is present, humans might value these animals less [75]. However, there is greater awareness of species’ roles within the environment. Thus, it is not needed to like certain species to protect them if humans are aware of animal value in the environment. The protection of species goes beyond our personal preferences under zoocentric, biocentric, and ecocentric perspectives.

From an anthropocentric point of view, one of the behaviors considered normal is that humans assess (or try to assess) their surroundings in terms of money. Money appeared as a human need to trade in order to survive. A theory about trade claims that trade was a “way of saving humanity from extinction” [76] and then became a source of safety and power. Thousands of years of “monetary” thinking are difficult to change. Money has become the benchmark for almost everything. For example, animal health value, especially livestock, is usually only a concern when it represents an economic loss. Only within anthropocentrism is monetary value important. However, for zoocentrism, biocentrism, and ecocentrism, money is not essential but represents a mean. These theories have more complexities and movements, with some exceptions that make them difficult to generalize. 

Complexity-aware or integrative approaches were proposed to solve health problems, such as “One Health” [77]. Within a biocentric perspective, this approach requires humans to think and act differently when making decisions about health, questioning how to improve for a more equitable consideration of humans, animals, and the environment in a socio-ecological system. This respect or moral consideration for other living beings places us to think about non-human animals, and their value. 

Biocentrism [29,78] and ecocentrism [79] are similar theories, but the latter is broader and harder to apply. An abrupt change from anthropocentrism [80] to biocentrism or ecocentrism might not be possible since it implies a more considerable mindset change. However, this transformation of thinking can be gradual and flexible since radical ways of thinking might not be healthy for anyone. For that reason, a perspective from the “anthropozoocentric interface” is proposed through which humans are respectful and flexible according to the situation in order to minimize the damage that they can cause through their decisions. 

The “anthropozoocentric interface” represents a friendly transition that can contribute to the “One Health” concepts. To protect species, healthy populations are needed, and for that, keeping the balance of human interference and non-interference in nature is a challenge [81,82,83,84]. A factor that may help is measuring the value of each living being and non-living things beyond money, prioritizing the point of narrower human–animal contacts and conflicts. “One Health” and “Well-being economy” [17] approaches are playing an important role to go beyond money for decision making in laws and regulations that affect health, well-being, and welfare. 

Worldwide, most countries are aware of animal suffering; thus, they have laws against animal cruelty, according to the Animal Protection Index (API) [28]. However, only a few countries recognize animal sentience, this being the ability of animals to feel and experience positive and negative emotions (pleasure, joy, fear, and pain) [28,85]. The WOAH has also been developing the Animal Welfare international Standards since 2002 [86], and animal welfare organizations have proposed to the United Nations the adoption of the “Universal Declaration on Animal Welfare” (UDAW), currently still in draft. 

Animal welfare awareness has increased over the years but as an isolated field. So, the challenge is to apply it in other fields, namely, health and laws. There is a lack of specific procedures such as quantitative representations of comprehensive animal value in public health laws and their integration into risk assessments of disease outbreaks. For example, the animal health regulation adopted by European Union in 2016, but applicable from 2021, includes only animal-transmissible diseases. This is the “Regulation 2016/429”, encompassing rules for the prevention, control, and eradication of disease through the traceability of animals and their products. This regulation does not include animal welfare but recognizes a connection between animal health and welfare [87]. This means that animal welfare is not mandatory for combating diseases, so non-monetary metrics to include animal health value or their welfare are not considered. For example, how governments proceed in case of disease outbreaks still arises controversies, such as mass culling of animals, for instance, in the case of 17 million minks killed for COVID-19 prevention in the absence of sufficient evidence that minks transmit the virus [88]. Is this rational? Would this still be an option in the future during disease outbreaks? Part of the answers depends on how societies perceive animals and how laws can be legislated based on that. 

Regarding animal welfare, some animal metrics were introduced, such as the *Welfare-Adjusted Life Years* (WALY) [89] to estimate the animal disease impact, including their welfare compromise. However, it only encompasses the animal factor without considering the human component. By contrast, the *Zoonosis Disability Adjusted Life Years* (zDALY) [22] integrates human and animal health, and this metric can be improved through the inclusion of more values proposed by this framework in order to be more comprehensive regarding different factors and species. Even though this framework seeks a more comprehensive approach, the environmental factor is still not represented through this metric and should be considered in the future. 

For most (anthropocentric) people [24], animal health only matters when it affects us, but having this mindset will not allow us to solve the underlying problems. The value of animals, as well as the value of human health, are difficult to translate in monetary terms. For that reason, concrete alternatives are needed for reshaping our thoughts, morality, and ethics under the scope of normative and environmental ethics. One of the reasons that environmental ethics is becoming valuable is because it is believed that the planet can survive without humans, but no humans can survive without the planet [90], leading us to think beyond us to find solutions out of anthropocentrism.

Regarding the health of populations, more qualitative research has been performed on humans [52,91] compared to animals; thus, human health might work as a benchmark to measure the health of other living beings. From the anthropocentric perspective [24], it is unacceptable to compare human health to animal health because under this perspective, humans feel superior, and animals are considered instrumental values, “something” that they can benefit from. From an anthropocentric perspective, humans can say that they care about animals or some of them, as long as they do not interfere with their own benefits, such as taking away part of their health or part of their lifetime. This creates a paradox that this framework calls the “animal paradox,” where humans affirm that they care about animals, but they do not in practice, especially if animals interfere with their interests and benefits. The “animal paradox” can also appear as a defensive mechanism for difficult situations. For example, in the case of euthanasia, for humans, this is generally prohibited to preserve life at all costs, whereas, for animals, it is generally accepted that animals can be euthanized to alleviate pain and suffering [92,93]. The loss of a patient is not easy but considering that animal life is not as “important” as human life might alleviate the guilt of many veterinarians. Therefore, keeping an anthropocentric perspective on animals makes the lives of veterinarians less difficult.

Something similar to the “animal paradox” happens with meat-eaters and animal lovers; someone can love some animals but eat meat, creating the so-called “meat paradox” [94]. Thus, avoiding the “uncomfortable” topic of the value of animals is a way to prevent cognitive dissonance and moral disengagement, especially for meat-eaters [95] and anthropocentric animal keepers. Cognitive dissonance appears when we act against our beliefs and values. In the case of moral disengagement, we convince ourselves that we are the exception to ethical standards in particular situations, such as the ones for the “animal paradox” and “meat paradox.” Veganism [96] appears as a zoocentric option for those who avoid the “meat paradox”; this philosophy under animal rights rejects the consumption of animals or their subproducts. Even though it has been a behavior change regarding certain topics, anthropocentrism has been the main approach to our decisions, and going against what we were taught our whole life can create an internal conflict that some people prefer to avoid. However, problems will not be solved by avoiding them but through deep and thoughtful thinking as a first step. New ways to value animals and their health can create new paths towards a “paradigm shift” [97].

Anthropocentrism has been the predominant way of thinking, especially in Western cultures and their religions [98]. Whereas Eastern religions such as Hinduism, Jainism, and Buddhism have in common a wider approach regarding the value of other species, similar to biocentric principles—even though biocentrism is not related to any religion [29]. Each person and culture have an inherent bias which is reducible by diversifying opinions. The more diverse the sources, the closer society will be to assessing the value of animals and learning more about ourselves. Theories cannot be debunked based only on surveys, but this helps clarify the perspectives of the participants to start from a different approach and continue debating about the value of animals for integrative solutions in health. Methods of dialogue and reflection should be complemented [99], but for that, this framework might facilitate a starting point. By collecting the values of many people from different backgrounds, it is expected to gain a better understanding of how humans value animals to contribute to “One Health” in order to address health challenges, such as zoonoses. 

Humans and animals share the environment, emotions, and potential diseases. Therefore, all decisions about animals affect humans, directly and indirectly. If animal health only matters when it affects humans, the anthropocentric perspective still continues, not a truly “One Health” which is based on biocentrism. This framework might guide us to more integrative efforts from an “anthropozoocentric interface” towards biocentrism. We are uncertain if the “One Health” approach is the panacea of zoonotic diseases, but only with tangible actions will societies have the answer in the future.

## 5. Final Considerations

Animals are more than commodities, and our actions, decisions, and laws should reflect that. As humans, it is our responsibility to make fairer decisions for them and us.

Science and society need to seize their strength for a change. So, the real question is if societies are ready to apply alternative paradigms to actions beyond anthropocentrism. This framework might help better tackle the dilemmas related to animal diseases, public health, and welfare.

## Figures and Tables

**Figure 1 animals-12-01845-f001:**
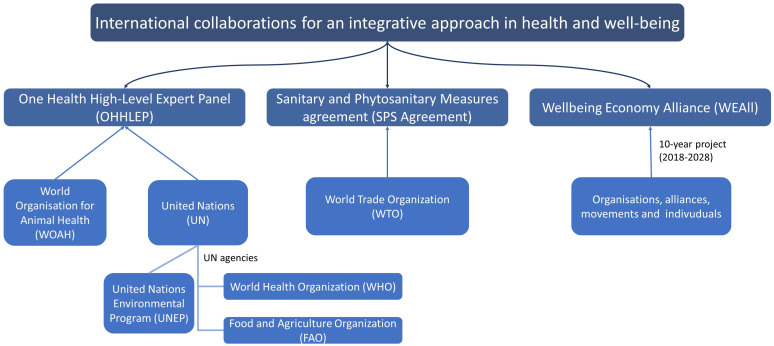
International collaborations that have recently addressed challenges in health with an integrative approach.

**Figure 2 animals-12-01845-f002:**
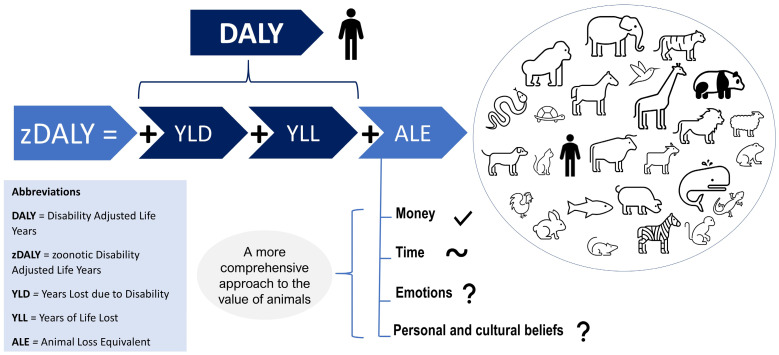
The zDALY: missing factors.

**Figure 3 animals-12-01845-f003:**
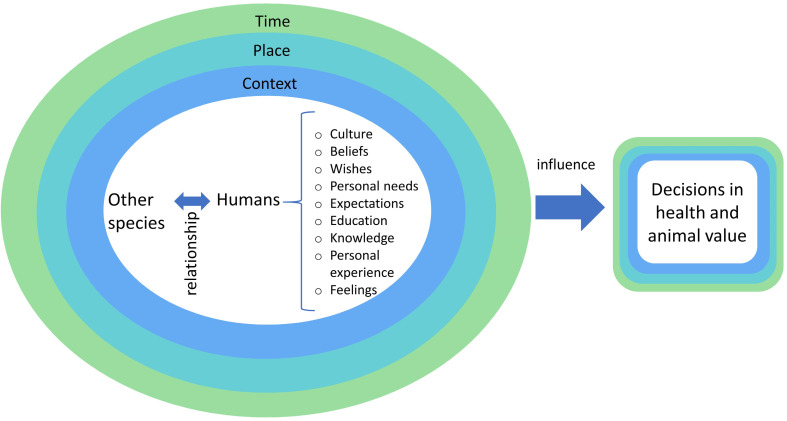
Factors that influence the way we value animals.

**Figure 4 animals-12-01845-f004:**
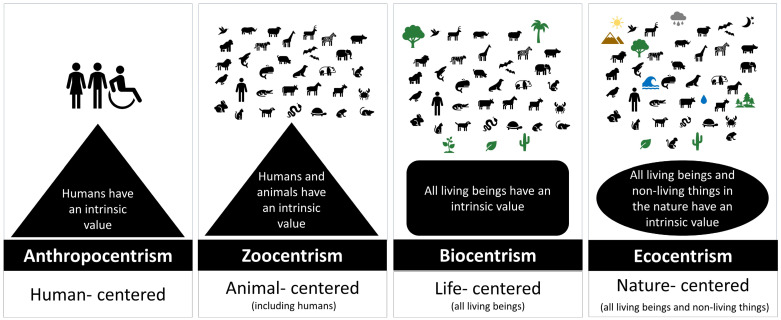
Main perspectives of environmental ethics to better understand the value of animals and health.

**Figure 5 animals-12-01845-f005:**
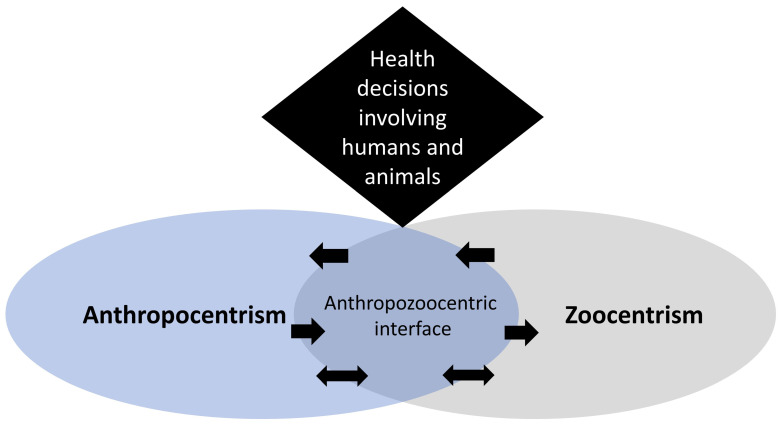
Anthropozoocentric interface: the direction of arrows indicates how the point of view varies according to specific situations (not always anthropocentrism or zoocentrism, sometimes neither of them).

**Figure 6 animals-12-01845-f006:**
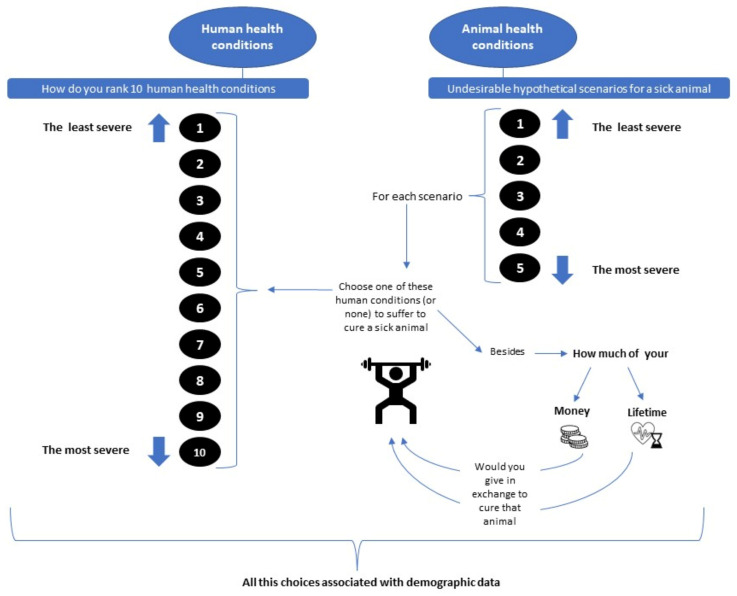
The employed methodology for animal health valuation.

**Figure 7 animals-12-01845-f007:**
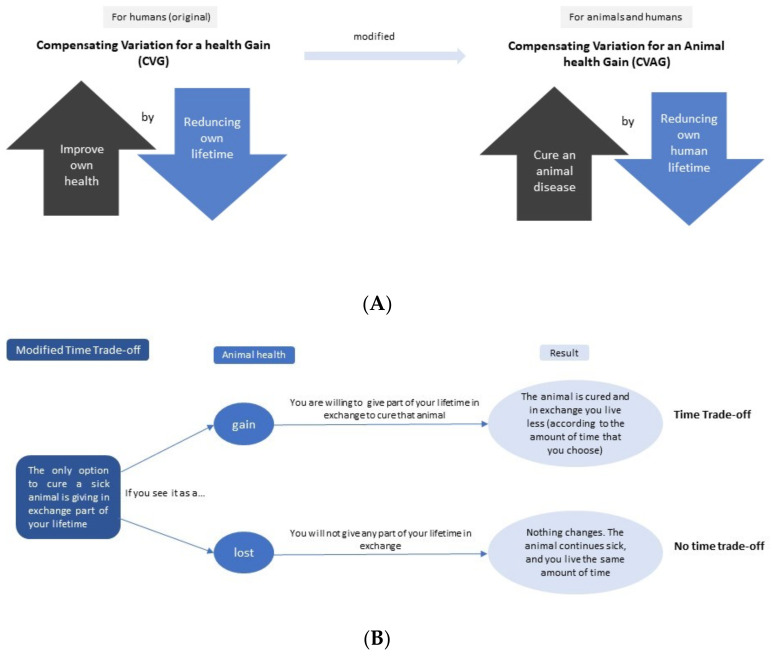
Animal Health Equivalence (AHE): methods for estimating the AHE in the zDALY. (**A**) Compensating Variation for an Animal Health Gain (CVAG). (**B**) Time trade-off according to the perception.

**Figure 8 animals-12-01845-f008:**
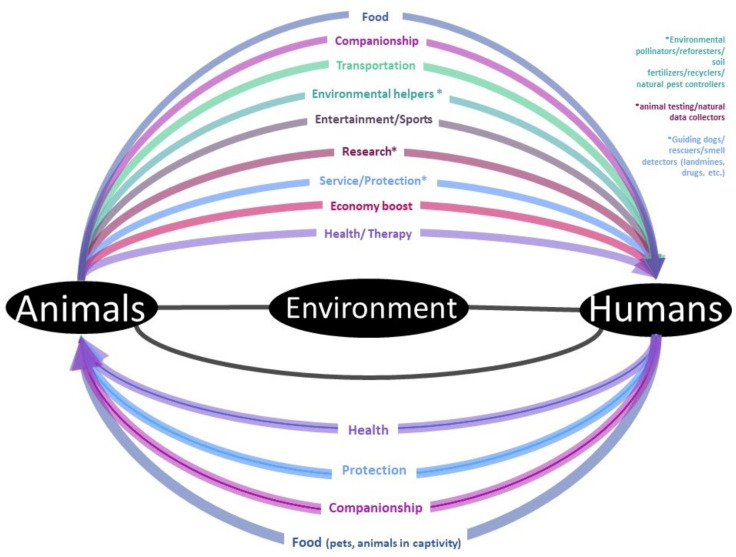
Human–animal relationship: a screenshot of a dynamic and complex interaction. The direction of the arrows indicates the beneficiaries in the interaction.

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
