# Peer review of "Alternative Paradigms in Animal Health Decisions: A Framework for Treating Animals Not Only as Commodities"

_animals, 2022, doi:10.3390/ani12141845_

Round 1

Reviewer 1 Report

This is an excellent thought provoking manuscript, including the references. My major suggestions for your consideration:

1) The title of your article includes "a framework for thinking beyond money", yet apparently this is so difficult that your main suggestion for the reader is to compare themselves to a sick animal and think about what they as an individual would be willing to trade to cure that sick animal, would they be willing to trade a certain amount of their time, their money and their personal health.  So, my suggestion is to be bolder, in the sense of how to treat animals as something other than commodities.

2) The discussion of ecocentrism; am I understanding this correctly, that ecocentrism believes that non-living things are equivalent to living things, that rocks should have the same rights as humans? And do you agree with this? If so, be more explicit that you do, if not say so.

A few trivial edits

1) line 36, add the word "and" i.e. war, and famine.

2) line 353: meat-eaters, not eat-meaters

Author Response

Dear Reviewer,

We appreciate your time and helpful comments. We have addressed your concerns to improve the manuscript. 

We have attached our answers.

Reviewer 2 Report

Dear authors, 

The text is very interesting, and leads us to a reflection on the subject.

I present some suggestions:

Abstract – The abstract does not present the real content of the work, it needs to be redone and the writing needs to be in the third person, as well as the entire text.

Figure 5 – is expendable.

Line 178 - “directly and directly” – Fix or replace

Line 212 - number the Discussion item (it would be 3)

Line 257 -263 – needs a reference

Line 281 – 287 and 288 – 295 - need references

Line 336 – 352 – Large paragraph, without a reference, please reference

Line 383 – 397 – Paragraph without reference and written in a completely personal way, the text is written in the first person and must be changed to the third person.

Add an item: 4. Final considerations and in this way a part of the discussion would be directed to “Final considerations”

The text is very interesting, and leads us to a reflection on the subject, but it lacks references to support the discussion.

The figures are self explanatory.

An English review must be done.

Author Response

(The authors gave the same response as above.)

Reviewer 3 Report

Summary

This study sets out an approach to incorporating alternative and additional valuations of animals with a view to use these for disease prevention decision making. It’s key and most relevant points made are around the differences in how animals are viewed and therefore valued according to species, culture, beliefs, place, time and context. The study proposes an interesting means to ascertain the value of animals using thought-provoking comparisons.

General comments

The paper is very logically presented, building on each concept systematically. However, evidence for statements made (in some cases very strong statements - are these just the authors’ opinions? If so, they should not be presented as fact) is lacking and strength of argument for choosing certain parameters and justifications throughout are weak, lacking in references / citations in many places.

The research aim – to consider the value of animals in ways other than monetary terms – is addressed with the proposals made for the framework. Justification for the specific parameters chosen i.e. comparison to human health conditions, money willing to spend to health and lifetime willing to trade, is addressed in part but not completely. There is a good discussion about why these additional parameters and other similar to this need to be considered although again lacking in supporting evidence / evidence-based facts.

In summary, the authors present an interesting idea but there is too much author opinion and too little evidence-based decision making throughout.

There are frequent grammatical / language errors, making it very challenging to read and follow in places.

Some specific comments

Introduction

Line 28 – sentence starts with “we”. Would be good to be more specific about who you are referring to or change to “There are challenges..”

Background

Line 32 – same comment as above – “We witness..” – who do you mean specifically?

Line 37 – and it could worsen – what could worsen? Frequency of outbreaks, severity of outbreaks? Not clear or specific enough.

Line 43 – First sentence grammatical issue ( see comments below) and does not make sense – what exactly are you trying to say here?

Line 44 – main component of what? Not clear or specific enough. Also, which international organisations? Provide examples.

Lines 56-58 – this sentence does not make sense, grammatical errors and difficult for readers to decipher the point being made. Re-write.

Lines 69 – 71 – why different font size??

Lines 88-89 – reference to back up this claim

Lines 89-90 – what is convenient?

Lines 92-93 –Refs to back up that this is a controversial topic?

Framework on alternative paradigms

A much better section.

Line 107 – ref for ecocentrism.

Lines 113-114 – ref.

Anthropozoocentric interface

Lines 134-135 – Refs?

Lines 143-145 – refs?

Example of application

Lines 251 -252 – no evidence for this claim, refs needed.

Lines 257 – 258 – refs to back up these aversions?

Lines 258-259 – refs again to back up your claims.

Discussion

Line 303 – Animal welfare has increased? Do you mean has improved? Or awareness of it has increased? Not clear.

Lines 304 – 306 – what do you mean when you say “no concrete actions”? What are concrete actions?

Line 328 – refs to back up your claim about most people’s view

Lines 332 – 335 – no refs to back up these claims

Lines 336-337 – no refs

Lines 368-369 – refs?

Suggested grammatical corrections

Abstract

Line 11 – requires

Background

Line 36 – war and famine

Line 42 – Last sentence is not a complete sentence and needs expanding into a complete sentence or amalgamating with the previous.

Line 43 – First sentence grammatically incorrect – needs to be re-written.

Line 61 – “lately” suggest change to “that have recently addressed…”

Line 63 – Metrics have been created to…

Line 89 – This is also…grammatical incorrect – It is also

Lines 92-93 – Re-write e.g. How we translate….in monetary term is a controversial topic.

Framework on alternative paradigms

Line 109 – “being this latter” not grammatically correct, neds re-writing.

Anthropozoocentric interface

Line 134 – choose

Lines 153 – 155 – rewrite this sentence, does not make sense in current format.

Lines 160 – this being

Lines 162 -163 – this sentence not clear, needs rewriting.

Example of application

Lines 178 – directly and indirectly?

Lines 179-180 – not a standalone sentence, more like a bullet point / instruction.

Lines 198 - -199 – not a standalone sentence

Discussion

Lines 234-235 – not a standalone sentence / incomplete sentence

Line 251 – which rather than with; strongly rather than stronger

Line 252 – connected to

Lines 259 – 260 – rewrite this sentence– that there is greater awareness of species’ roles within the environment.

Line 302 – remove being from in front of currently

Lines 304 – 306 – rewrite this sentence, grammatically incorrect.

Line 314 – still results in controversies

Line 331 – to call concrete alternatives

Lines 371-372 – rewrite, does not make sense

Lines 385-386 – one health approach

Lines 390-391- sentence needs rewriting, grammatically in correct

Lines 394-395 – Need to rewrite this sentence

Lines 395-397 – is are societies ready for

Lines 396 – change concrete actions – not correct use of these words

Author Response

(The authors gave the same response as above.)

Round 2

Reviewer 1 Report

Thank you for taking my suggested theoretical issue to heart and extensively revising the manuscript accordingly, this new version is excellent. I look forward to seeing it published!